# Disaggregation of Green Space Access, Walkability, and Behavioral Risk Factor Data for Precise Estimation of Local Population Characteristics

**DOI:** 10.3390/ijerph21060771

**Published:** 2024-06-14

**Authors:** Saurav Guha, Michael Alonzo, Pierre Goovaerts, LuAnn L. Brink, Meghana Ray, Todd Bear, Saumyadipta Pyne

**Affiliations:** 1Health Analytics Network, Pittsburgh, PA 15237, USA; 2Department of Statistics, Mathematics & Computer Application, Bihar Agricultural University, Bhagalpur 813210, India; saurav@bausabour.ac.in; 3Department of Environmental Science, American University, Washington, DC 20016, USA; alonzo@american.edu; 4Biomedware, Inc., Ann Arbor, MI 48103, USA; goovaerts.pierre@gmail.com; 5Allegheny County Health Department, Pittsburgh, PA 15219, USA; lbrink@achd.net; 6Heed Lab, North Bethesda, MD 20723, USA; 7Department of Family Medicine, School of Medicine, University of Pittsburgh, Pittsburgh, PA 15260, USA; 8Department of Statistics and Applied Probability, University of California, Santa Barbara, CA 93106, USA

**Keywords:** small area estimation, built environment, GIS, kriging, zip code, Allegheny County

## Abstract

Background: Social and Environmental Determinants of Health (SEDH) provide us with a conceptual framework to gain insights into possible associations among different human behaviors and the corresponding health outcomes that take place often in and around complex built environments. Developing better built environments requires an understanding of those aspects of a community that are most likely to have a measurable impact on the target SEDH. Yet data on local characteristics at suitable spatial scales are often unavailable. We aim to address this issue by application of different data disaggregation methods. Methods: We applied different approaches to data disaggregation to obtain small area estimates of key behavioral risk factors, as well as geospatial measures of green space access and walkability for each zip code of Allegheny County in southwestern Pennsylvania. Results: Tables and maps of local characteristics revealed their overall spatial distribution along with disparities therein across the county. While the top ranked zip codes by behavioral estimates generally have higher than the county’s median individual income, this does not lead them to have higher than its median green space access or walkability. Conclusion: We demonstrated the utility of data disaggregation for addressing complex questions involving community-specific behavioral attributes and built environments with precision and rigor, which is especially useful for a diverse population. Thus, different types of data, when comparable at a common local scale, can provide key integrative insights for researchers and policymakers.

## 1. Introduction

National discussions of health and health disparities in the United States (US) have placed a spotlight on the social determinants of health (SDH) [1,2]. SDHs are defined by the WHO as “the conditions in which people are born, grow, live, work, and age” [3]. Health disparities are generally embedded in social or structural determinants of health, which cannot be effectively addressed at the individual level alone [4]. Social characteristics, such as race/ethnicity and socioeconomic status, are interconnected with key influences from physical environments, access to health care services, and psycho-social experiences rooted in socio-environmental factors [5]. In recent years, environmental contributions to health outcomes have received increasing attention. The WHO, for instance, estimates that 13–32% of the global disease burden is attributable to environmental determinants, including exposure to pollution and chemicals, physical exposures, the built environment, and other anthropogenic changes [6].

Extension of the SDH to Social and Environmental Determinants of Health (SEDH) provides us with a conceptual framework to gain insights into possible associations among different human behaviors and the corresponding health outcomes that take place often in and around complex built environments. Neighborhood, for example, serves as a source of SEDH which can be viewed through its components of the built environment, services, the residents and their interactions [7]. Higher rates of obesity in neighborhoods with poor walkability, low access to healthy foods, lack of health care facilities, are some of the mechanisms through which built environments can influence health outcomes. These may include mortality, life expectancy, mental health, as well as non-communicable diseases such as obesity and diabetes [8,9]. Conversely, the extant literature on public health, environmental psychology, and other disciplines notes that neighborhood environments that are conducive to walking and exercising (including presence of sidewalks, trails and parks, recreational facilities within walking distance), beautiful scenery, and green surroundings allow residents in those neighborhoods to avoid sedentary lifestyles, reduce reliance on automobiles, and be more physically active [10,11,12,13].

Developing better built environments requires an understanding of those aspects that are most likely to have a measurable impact on the target SEDH. In terms of health and weight, neighborhood features of influence include walkability [14,15], proximity to parks and trails [16,17], and access to different means of physical activity [18]. While several observational studies have explored the impact of built environments on weight change, weight gain, and obesity, it is important to think about additional characteristics of a neighborhood that may either enable or inhibit individual decisions to be physically active. For instance, beautiful scenery and green surroundings serve to enrich one’s walking experience just as maintained and well-lit streets may promote feelings of safety, thus enhancing walkability in the neighbourhood. In fact, research indicates that perceived neighborhood walkability influences physical exercise independent of individual preferences [19,20].

Availability of green spaces and walkable neighborhoods may promote social cohesion by providing meeting places, along with increased perceptions of familiarity and safety that can, in turn, promote physical activity. It is a reasonable expectation that a person would experience better health if they lived in an area where they are physically and socially active, feel supported, and safe, and trust their neighbors. On the other hand, neighborhood characteristics could have an independent effect on unhealthy behaviors, such as tobacco use, alcohol, and smoking [21,22]. For instance, distressed physical features of neighborhoods (e.g., poor quality housing, litter, damage) and inadequate basic resources (such as inadequate public transportation and recreational facilities) are considered to be important determinants of smoking and drinking behaviors [23]. Neighborhoods can also serve as vehicles of residential segregation and limited access to employment opportunities that could lead to disparities in health and well-being. This can, in turn, manifest as decaying commercial corridors saturated with liquor stores, dilapidated buildings, and infrastructure that contribute to health disparities through myriad pathways, including crime, injury, depression, addiction, etc. [24].

In recent years, research has focused on combining health and behavioral risk data with built environment data. Aggregated health data are often provided at different spatial levels—from county and metropolitan areas to zip codes, census tracts, or nearest intersections [25]. Similarly, built environment data contains several variables of interest, again at different geospatial levels, thus making it necessary to develop new methods of spatial analysis to disaggregate the different types of data at different scales. Let us, for instance, take the well-known Behavioral Risk Factor Surveillance System (BRFSS) surveys [26] of the U.S. Centers for Disease Control and Prevention (CDC). Disaggregation of such population behavioral risk data to finer geographical scales can provide key insights into certain public health challenges that are uniquely present at spatially local levels, such as municipalities or zip codes, where the effects of a given SEDH of interest may be most precisely understood, and thus, effectively actionable.

We focused on the problem of estimating multiple behavioral risk factors, along with green space access and walkability, in Allegheny County, in southwestern Pennsylvania (PA). The county is also a part of the multistate Appalachian region and is centered around the city of Pittsburgh. It is home to 1.25+ million people in the 2020 census, thus making it the second largest county by population in the state (PA). Interestingly, the mortality rate in Allegheny County is higher than those of both the state of PA and the U.S. For example, the 2018 age-adjusted mortality rate (765.5 per 100,000) was higher than both PA (759.4 per 100,000) and the U.S. (723.6 per 100,000) [27]. The county conducts the Allegheny County Health Surveys (ACHS), which are modeled after the CDC BRFSS surveys. In the past, such surveys as BRFSS have produced longitudinal measures of key risk factors and useful indices on social vulnerabilities in the population. In particular, studies using ACHS have focused on the estimates and disparities of various behavioral risk factors that exist in Allegheny County [28,29,30].

We aim to extend past studies by employing different techniques to fulfil different purposes of data disaggregation. Often, sample surveys have limited coverage of small geographic areas, such as municipalities or zip codes, which could make estimation of local population characteristics (e.g., SEDH) for such small areas unreliable. To address this issue, we adopted a model-based Small Area Estimation (or SAE) approach on a prominent survey (ACHS), by leveraging on auxiliary socioeconomic information about the local populations, which yielded reliable estimates. The validated results were used for ranking all the zip codes of Allegheny County and providing insights into three well-known health-related risk behaviors of the corresponding populations. Further, we used GIS data processing to determine access to green spaces by the residents within each zip code of Allegheny County. Finally, an area-to-point (ATP) kriging method was applied for geostatistical estimation of zip code-specific walkability measures within the same county. Data on behavioral risk factors, green space access, and walkability, upon rigorous disaggregation at the zip code level, allows us to perform mapping, ranking, and more precise studies of SEDH of particular interest in the underlying population.

## 2. Materials and Methods

### 2.1. Data

In this study, we conducted secondary data analysis of Allegheny County Health Survey (ACHS) [28] as described below.

*Behavioral Risk Factor Survey data:* The Allegheny County Health Department conducted in 2015 a local survey (ACHS), which was modeled after the CDC’s BRFSS. ACHS asked the standard BRFSS questions and included additional questions of local relevance to Allegheny County adults (ages 18 and over). See further details of the survey administration in [28,29]. A total of 9032 responses were collected, analyzed, and reported, of which 74 responses were excluded due to error in ages and 122 excluded due to missing zip codes, thereby providing a total of 8836 responses across 105 zip codes. Demographic variables were then re-categorized to be consistent with the census categories. (See Appendix A for re-categorized sociodemographic profiles of survey respondents). For this study, we considered three health risk related variables, body mass index (or BMI), smoking, and moderate physical activity. All these variables were converted to binary variables for estimating the proportion of population with greater health risk in Allegheny County. BMI (kg/m^2^) value of 25 or above indicated overweight or obesity, and thus we assigned 1 if an individual has BMI ≥ 25, and 0, otherwise. Next, if an individual has smoked at least 100 cigarettes in his/her life, then the variable ‘Ever-smoker’ is assigned to 1, and 0, otherwise. The third variable ‘Moderate Activity’ is assigned 1 if the total duration of the activities performed by an individual is less than 150 min in a week, and 0, otherwise. Such moderate activities include brisk walking, gardening, etc. Four socioeconomic auxiliary variables provide local (zip code tabulation areas), i.e., contextual, information for Small Area Estimation in this study: the percentage of families in poverty, the proportion of population above 65 years old, the proportion of blacks in the population, and the proportion of adults with a bachelor’s degree or higher qualification.

*Green spaces data:* The Allegheny County Greenways data for 2021 [31] were obtained from Pennsylvania Spatial Data Access (PASDA, which is PA’s official public access open geospatial data portal). The Greenways feature class consists of a compilation of GIS data (with lat–long coordinates) on various categories of parks, trails, land green properties, park nodes, and City of Pittsburgh Greenways, which we included among accessible green spaces. Other categories, such as agricultural easements, sensitive slopes, etc., were excluded from our analysis. (A summary is available in Appendix A.)

*Walkability data:* We obtained the 2010 walkability ratio data for the census tracts of Allegheny County from the Western Pennsylvania Regional Data Center [32] (WPRDC, which is managed by the University of Pittsburgh Center for Urban and Social Research). It provides the ratio of the length of sidewalks in a census tract to that of its streets as a measure of pedestrian infrastructure. Note that this ratio could go up to 2, since both sides of a road could have sidewalks. Further details about the Walkability ratio appear on the WPRDC website [32].

### 2.2. Methodology

#### 2.2.1. Small Area Estimation of Behavioral Risk Factors in a Zip Code

We used the model-based Small Area Estimation (SAE) method to obtain zip code-specific estimates of 3 behavioral risk factors: *Body Mass Index* (BMI), *Ever-smoking*, and *Moderate activity*. These variables were captured by the sample survey (ACHS) described above. Let us assume that a finite population of interest Ω of size *N* and a sample *s* of size *n* is drawn from Ω with a given sampling design. The population Ω is assumed to consist of *D* small areas (hereafter referred to as “areas”). The population units in the area d=1,…,D is represented by $\Omega_{d}$, with known population size $N_{d}$, such that $\Omega ={⋃}_{d=1}^{D}\Omega_{d}$ and $N=\sum_{d=1}^{D}N_{d}$. Here, we used the subscript *s* and *r* to denote the units belonging to the sample and non-sampled parts in the population, along with $n_{d}$ denoting the sample size in area *d*. For area *d*, the units making up the sample are denoted by $s_{d}$ such that $s={⋃}_{d=1}^{D}s_{d}$ and $n=\sum_{d=1}^{D}n_{d}$. Let $y_{dj}$ be the value of a study variable *y* for unit *j* in area *d*. The target variable is assumed to be a ‘counting’ variable, i.e., it takes values in the set of non-negative integers (with binary as a special case) and the objective is to estimate the small area population counts $y_{d}=\sum_{j\in \Omega_{d}}y_{dj}$ or the small area proportions $P_{d}=N_{d}^{-1}\left(\sum_{j\in \Omega_{d}}y_{dj}\right)$.

We denote the $y_{sd}=\sum_{j\in s_{d}}y_{dj}$ as the sample count in area *d*. Area-specific *p*-vector auxiliary variables denoted by $x_{d}$ is available from secondary data sources (e.g., the census or administrative registers). A *D*-vector area-specific random effects is denoted by $u={\left(u_{1},\ldots ,u_{d}\right)}^{T}$, where $u_{d}∼N\left(0,\sigma_{u}^{2}\right)$. The target variable $y_{d}$, conditional on the random area effect $u_{d}$, is then assumed to follow an exponential family of distributions, i.e., $y_{d}\left|u_{d}∼\right.f(\eta_{d})$, where $\eta_{d}$ is a function of $u_{d}$ and
(1)$$f\left(\eta_{d}\right)=exp\left[\frac{y_{d}\eta_{d}-a\left(\eta_{d}\right)}{\theta_{d}}\right]+b\left(y_{d},\theta_{d}\right).$$

The distribution $f\left(\eta_{d}\right)$ is characterized by the canonical parameter $\eta_{d}$, a known scaling parameter $\theta_{d}$ and functions a(·) and b(·). As the target is to model the counts, we concentrate on the Poisson and Binomial distributions for which the parameter $\eta_{d}$ is expected to follow a linear mixed model. According to Johnson et al. (2010) [33] and Chandra et al. (2011) [34], when the objective is proportion estimation for the small areas, the sample count is
$$y_{sd}\sim Bin(n_{d},\pi_{d}),$$
where $\pi_{d}$ is the probability of prevalence in area *d*, (d=1,…,D). The GLMM with logistic model linking $\pi_{d}$ with the covariates $x_{d}$ is then
(2)$$logit\left(\pi_{d}\right)=\ln\left\{\pi_{d}{\left({1-\pi }_{d}\right)}^{-1}\right\}=\eta_{d}=x_{d}^{T}\beta +u_{d}$$
where $\pi_{d}=\exp\left(\eta_{d}\right){\left\{1+\exp(\eta_{d})\right\}}^{-1}=expit\left(\eta_{d}\right)$.

Under this model, the mean of $y_{sd}$ given $u_{d}$ is
(3)$$\mu_{sd}=E\left(\left.y_{sd}\right|u_{d}\right)=\eta_{d}expit\left(x_{d}^{T}\beta +u_{d}\right).$$

Without loss of generality and putting $\pi ={\left(\pi_{1},\ldots ,\pi_{D}\right)}^{T}$ and $X={\left(x_{1}^{T},\ldots ,x_{D}^{T}\right)}^{T}$, aggregating different area level models leads to the population level model as
(4)$$g\left(\pi \right)=\eta =X\beta +u$$
where $g\left(\pi \right)={\left(g\left(\pi_{1}\right),\ldots ,g\left(\pi_{D}\right)\right)}^{T}$ and $\eta ={{(\eta }_{1},\ldots ,\eta_{D})}^{T}$. A plug-in empirical predictor (EP) of the population count $y_{d}$ in area *d* is
(5)$${\hat{y}}_{d}^{EP}=y_{sd}+{\hat{\mu }}_{rd}=y_{sd}+(N_{d}-n_{d}){\hat{\pi }}_{d}^{EP}$$
where ${\hat{\pi }}_{d}^{EP}=expit({\hat{\eta }}_{d})$ for a binary target variable with ${\hat{\eta }}_{d}=x_{d}^{T}\hat{\beta }+{\hat{u}}_{d}$.

An estimate of the proportion in area *d* is given by ${\hat{P}}_{d}^{EP}=N_{d}^{-1}{\hat{y}}_{d}^{EP}$. For non-sampled areas (i.e., areas with zero sample sizes), synthetic estimation is the usual approach to estimate the proportions or counts [34]. For non-sampled area *d.out*, the synthetic estimate of $\eta_{d}$ is, ${\hat{\eta }}_{d,out}=x_{d,out}^{T}\hat{\beta }$, where $x_{d,out}$ is the vector of covariates for non-sampled areas and the synthetic estimator of $y_{d}$ is given by ${\hat{y}}_{d}^{SYN}=N_{d}{\hat{\pi }}_{d}^{SYN}$, with ${\hat{\pi }}_{d}^{SYN}=expit({\hat{\eta }}_{d,out})$ for a binary target variable.

*Model diagnostics:* Two kinds of diagnostic measures are generally used in small area estimation, viz., the model diagnostics and the small area diagnostics [35,36]. The model diagnostics are useful to validate the underlying model assumptions while the small area diagnostics specify the validity and reliability of the estimates. The random area effects (here, zip code level) are expected to follow a normal distribution with zero mean and constant variance. If the model assumptions are fulfilled, then the zip code level residuals are likely to be distributed randomly around zero. Scatterplots, histograms and q–q plots are used to examine these model assumptions.

*Wald’s test of Goodness-of-Fit (GoF):* For the sample zip codes, GoF was applied to examine whether there is a significant difference between the direct and the model-based small area estimates. With the assumption that the direct and the model based small area estimates are distributed independently, which is quite reasonable for large sample sizes, the value of the GoF test statistic can be equated with a suitable critical value from a chi square distribution, with degrees of freedom (*D*) equal to the total number of sample zip codes.

*Small area diagnostics:* We have used two widely accepted SAE diagnostic measures, (a) the bias diagnostic and (b) the percentage coefficient of variation (%CV) diagnostic [37], to assess the validity and the reliability of the model-based small area estimates of the target variables, BMI, Ever-smoker and Moderate activity. The former diagnostic measure assesses the validity while the second determines the precision of the model-based estimates. The elementary idea behind the bias diagnostic measure is that, since direct estimates are unbiased, the regression of direct estimates on the true values should be linear and correspond to the identity (y = x) line. If the model-based estimates are “close” enough to these true values, then the regression of the direct estimates on these model-based estimates should be similar to the identity line. Hence, we plotted the direct estimates (in Y-axis) against the corresponding model-based small area estimates (in X-axis) and searched for departure of the fitted least squares regression line (shown by dashed line) from the Y = X line (solid line). Moreover, the small area estimates with low %CVs are considered to be precise and reliable.

#### 2.2.2. Geospatial Estimation of Access to Green Spaces in a Zip Code

We conducted step-by-step analysis of geospatial data to quantify individual access to green spaces in each zip code within Allegheny County. Within our 126 zip code polygons, we sought to estimate the residents’ access to green spaces that may reasonably be used by the public. Typically, such uses of green spaces are for recreation and/or exercise as noted in Appendix A. These green spaces were selected from the Allegheny County Greenways dataset (containing a total of 53,035 polygons over 5787 km^2^) leading to a final green space count of 1210 polygons with an area of 1296 km^2^.

For estimating the *Mean bin_green* of a zip code *Z*, which is defined by the proportion of the population residing in *Z* that has access to any green space within 500 m, the steps of our algorithm proceeded as follows:Select “accessible” subset of green spaces (e.g., exclude golf courses or sensitive slope areas)Generate spatially random sample points within census block polygons with amounts proportional to block population (n = 150,000). These points are meant to plausibly represent the location of individuals within the population.Exclude points that were placed within the green spaces themselves or within 500 m of the study area boundary.Randomly retain 10% of the data for further analysis (n = 15,000)Sum the area of accessible green space within 500 m of each sample point. This step also results in a count of nominally unique green spaces within each buffer.At each sample point, calculate the percent of the buffer occupied by green space (i.e., a continuous variable showing green space amount rather than binary accessibility only).Summarize the results of Step 6 to the zip code level.

In this study, a resident’s accessibility to green space is defined as living in a census block within 500 m of the edge of an eligible green space. Here, after weighting for census block population, we generated 150,000 spatially random points within the Allegheny County census blocks. These points represented plausible residential locations for our sample population. After generating the points, we then filtered out those lying inside green spaces, and those within 500 m of the County boundary. Finally, we randomly selected 10% (i.e., 14,090) of the filtered samples to retain for our analysis.

To quantify the amount of green space accessible to each “person” (as represented by a sample point), we summarized the green space inside of 500 m buffers around each location. Summarizing by random sample point location rather than by the zip code polygon itself is important because it alleviates boundary issues, i.e., it allows for the accessible green space for any person to be in either their own or a neighboring zip code. Once each sample point was associated with the green space access information, the zip code polygons were used to compute the proportion of the population residing in a zip code that has access to any green space within 500 m as measured by the *Mean bin_green* variable.

#### 2.2.3. Geostatistical Estimation of Walkability Ratio in a Zip Code

The *Mean walkability ratio* for each zip code was computed by data disaggregation via a geostatistical interpolation technique known as area-to-point (ATP) kriging [38]. Based on walkability ratio data from 402 census tracts (hereafter referred to as tract), the Mean walkability ratio of each Allegheny County zip code was estimated by the following spatial interpolation approach:The CT data underwent a normal-score transform [39] to attenuate the impact of a few extreme values on the subsequent analysis.The semi-variogram of normal scores was calculated and modeled using the iterative procedure described in Goovaerts (2008) [40].The normal scores were estimated at the nodes of a 200 ft. spacing grid overlaid over all 125 Allegheny County zip codes. The estimation was conducted using ATP kriging, and the 12 closest tract data located within a 33,000 ft. radius of each grid node. This radius corresponds to the range of autocorrelation of the semi-variogram modeled in Step 2.The kriging estimates were normal-score back-transformed [39] to yield for each grid node an estimate of the walkability ratio.For each zip code, the Mean walkability ratio was calculated by averaging the kriging estimates for all grid nodes falling within that zip code.

Notably, because of the coherency constraint of ATP kriging [38], averaging the grid node estimates falling within each tract would return the original walkability ratio for that tract.

A flowchart describing the analytical pipeline of disaggregation and use of different types of data in this study is shown in Figure 1.

## 3. Results

GIS data from the Allegheny County–Greenways dataset was used not only to locate the green spaces all across the county but also to select the types that are most likely to allow public access, e.g., different types of parks in different areas (see Appendix A). Two types of statistics were computed—percent green space within 500 m and a binary value that indicates whether there is any green space within 500 m of a random individual’s location in a zip code. We used the latter *Mean bin_green* (defined by the proportion of the population residing in Z that has access to any green space within 500 m) as it better represents access to green spaces owing to the weighted selection of point locations by the underlying block populations The map of MBG (Figure 2A) highlights the overall spatial distribution along with the local disparities in green space access in the county.

The data on walkability ratios for Allegheny County were available at the level of census tracts. As we are interested in conducting an analysis of different SEDH variables at the zip code level, we performed side-scaling of the data by a geostatistical interpolation approach called area-to-point (ATP) kriging. Thus, the kriging estimates were backtransformed [39] to yield for each grid node, and the mean walkability ratio for each zip code was calculated by averaging the kriging estimates for all grid nodes that fall within that zip code (see Figure 2B).

A generalized linear model was fit with four auxiliary variables for small area estimation of three key behavioral risk factors with potential health outcomes, BMI, Ever-smoker and Moderate activity for each zip code in Allegheny County. The ACHS questionnaire directly targets the underlying population while the selected sociodemographic variables enrich our model with ancillary but spatially contextual information. As described below, we conducted a detailed array of diagnostics to test the model’s assumptions and goodness-of-fit as well as the bias and reliability of the resulting estimates. Once the estimates were found to be reliable and precise, we used these for ranking the zip codes by each of risk factors. Finally, the median of these ranks was computed for each zip code, which are listed in Appendix A.

Figure 3 displays the scatterplot (left), histograms (center), and normal q–q plots of the zip code level residuals (right) for the three target variables, BMI, Ever-smoker, and Moderate activity (from top to bottom respectively). These plots in Figure 3 reveal that the zip code level residuals are distributed randomly around zero. The histograms and the q–q plots offer enough evidence in support of the normality assumption. The Shapiro–Wilk test has also been used to examine the normality of the zip code specific random area effects.

The values of Shapiro–Wilk test statistics for the zip code level residuals, with 84, 90 and 88 degrees of freedom, respectively, for BMI, Ever-smoker, and Moderate activity, are 0.9846, 0.9749, and 0.9733, and corresponding *p*-values are 0.408, 0.093, and 0.063 for models fitted with BMI, Ever-smoker, and Moderate activity, respectively. In each scenario, the *p*-value of the Shapiro–Wilk test statistics was greater than 0.05, thus confirming the non-rejection of the null hypothesis that the zip code level residuals are normally distributed. The model diagnostics measures evidently reveal that the normality assumptions were fulfilled reasonably well by the data.

To test Goodness-of-Fit (GoF), the value of Wald test statistic was computed to be 85, 91 and 89 for BMI, Ever-smoker, and Moderate activity, respectively. These values correspond to critical values of 107.52, 114.26 and 112.02 at 5% level of significance. A Small value (<critical value) of the test statistic indicates no statistically significant difference between the model-based and the direct estimates. The values of the test statistic for the model-based estimates of BMI, Ever smoker, and Moderate activity are 1.021, 1.785 and 1.292, respectively. These values are smaller than their respective critical values, which indicates that the model-based small area estimates are consistent with the direct estimates. The bias diagnostic plots in Figure 4 clearly indicate that the model-based estimates of BMI, Ever-smoker, and Moderate activity are less extreme when compared to the corresponding direct estimates, demonstrating that the typical SAE outcome shrinks the extreme values towards the average. The values of adjusted R^2^ for the fitted regression line between the direct estimates and the model-based estimates for BMI, Ever-smoker, and Moderate activity are 99.01, 99.02 and 99.14 percent, respectively.

Finally, we compared the percent coefficient of variation of the small area estimates with direct estimates, since SAE with low CVs are considered precise and reliable. Distribution of %CVs of direct and small area estimates are reported in Table 1. Zip code-specific %CVs of direct, and small area estimates are given in Figure 5. Table 1 and Figure 5 show the unstable direct estimates of BMI, Ever-smoker, and Moderate activity; CV of BMI ranges from minimum of 4.93 to maximum of 90.20% (mean = 18.64%) while CV of Ever-smoker ranges from minimum of 5.99% to maximum of 100.01% (mean = 24.89%) and CV of Moderate activity ranges from minimum of 8.98% to maximum of 102.61% (mean = 32.22%). On the other hand, small area estimates seem to be reasonably stable as the CV of BMI ranges from minimum of 5.33% to maximum of 18.31% (mean = 9.63%), while CV of Ever smoker ranges from minimum of 6.63% to maximum of 32.82% (mean = 11.84%) and CV of Moderate activity ranges from minimum of 9.68% to maximum of 53.99% (mean = 18.18%). The model based small area estimates appear to be more precise and reliable than the direct estimates.

To take a closer look into the spatial distributions of the behavioral risk factors vis-à-vis greenspace access and walkability ratio at the zip code level, we produced bivariate choropleth maps of Allegheny County. Figure 6 top panel shows the local distribution of the greenspace access in Allegheny County against BMI, Ever-smoker, and Moderate activity, respectively, by zip code, while its bottom panel demonstrates the same for walkability ratios. Zip codes with higher risk factor estimates and poorer green space access (MBG) and walkability (MWR) are depicted in darker shades. Such (and more) local characteristics of the top 10 zip codes in Allegheny County as per Median rank by SAE, which have at least 100 housing units, are listed in Table 2. We noted that, while these zip codes have, in general, higher than median individual income for the county (slightly above $30,000), their median values of MBG and MWR are very similar to those of the entire county. The full list of the computed small area estimates of the behavioral risk factors of interest for each zip code in Allegheny County is given in Appendix A. Zip code level estimates of BMI (red), Ever-smoker (blue) and Moderate activity (green), and the median rank (orange) based on them, are mapped in Figure 7.

Disaggregation of data can lead to both new data-driven questions and explanations for existing areal disparities, as well as absence thereof (e.g., how could fundamentally distinct areas have comparable SEDH?). For example, based on at least the three estimated behavioral risk factors, we note that the boroughs of Braddock (zip 15104) and Tarentum (zip 15084) have similar Median ranks (15 and 16, respectively). Braddock is located in the eastern suburbs of Pittsburgh on the Monongahela River and its population is 16.3% white and 74.4% black. Tarentum, which lies northeast of the county along the Allegheny River, has an 85.1% white and 6.4% black population. Braddock and Tarentum may have key differences in racial composition, but they share similarities in their relatively high median ages of 40 and 45 years, respectively, which might explain their poor ranks in terms of Moderate activity (97 and 90). Interestingly, they share high ranks in both Ever-smoker (12 and 6) and BMI (15 and 16). While both boroughs may have comparable population sizes and number of housing units, Braddock has almost ten times the population density and half the median household income of Tarentum. Yet Braddock has notably higher walkability (1.028) and green space access (0.766) than Tarentum (correspondingly, 0.222 and 0.502). Such comparisons demonstrate the complexity of questions involving community-specific attributes and built environments that could be addressed quantitatively and with precision, which is especially useful for a diverse population.

## 4. Discussion

In this study, availability of data at a suitable spatial scale can help us take a closer look at the local characteristics of neighborhoods and shed light on their different built environments and behavioral outcomes. Indeed, in the U.S., the place of residence has emerged as a powerful SEDH in recent years, with some researchers suggesting that zip codes may have higher impact on health outcomes than even genetic codes [40]. Such suggestions are rooted in the effects of governmental policy, societal values, and cultural norms on the creation of communities (including redlining and segregation) and its implications on health risk factors and outcomes. The interest in the topic of “neighborhoods and health” has been driven by several interrelated trends within public health and epidemiology [41]. Beyond purely individual-centric determinants of disease [29,41,42,43], local areas shed light on relevant contexts that could possess both physical and social characteristics affecting the lifestyles (e.g., walkability), exposures (e.g., green spaces) and other parameters of health of their residents.

Current SEDH conceptual frameworks acknowledge that underlying indicators are important factors in differential health outcomes [44,45,46]. Several index-based approaches, such as the Area Deprivation Index [47], Social Vulnerability Index [48], etc., have emerged to serve as proxy measures of SEDH. There is general cognizance that minorities and low socioeconomic status populations disproportionately live in resource poor communities, and that these communities vary widely in terms of their geography and built environment, including constrained lifestyles (lack of physical spaces to exercise, food deserts, etc.), social stressors (e.g., violence) or psychosocial factors. The present study illustrates the application of disaggregation to different types of data—population surveys, GIS and environmental—which could be used to inform policymaking, such as design of interventions that are precise and targeted in mitigating disparities.

We note that our study has certain limitations. It is difficult to fit a single explanatory model to establish cause-and-effect or any direct connections across such complex contexts as built environments and behavioral risk factors. Other potential limitations of the study are due to the self-reporting based BRFSS data, and the longitudinal variations between data collected on built environments from different time periods. While the findings may be specific to the studied county, our demonstration indicates that similar approaches could be used for other geographical locations. In particular, research on built environments may be expensive, involving neighborhood audits that include costs and logistics for manual and detailed data collection. Thus, in recent years, several studies have set out to conduct geospatial analyses to evaluate the built environment. While precise estimation of disaggregated data on local characteristics, such as green space access or walkability, do not guarantee the lowering of any behavioral risk for the community, it does provide a quantitative approach for addressing the mechanistic interactions of individuals and groups with their distinctive environments that potentially determine important health outcomes.

## 5. Conclusions

Rigorous disaggregation of zip code-level data of Allegheny County in southwestern Pennsylvania produced tables and maps of the local characteristics, which revealed their overall spatial distribution along with disparities therein across the county. We noted that, while the top ranked zip codes by behavioral estimates generally have higher than the county’s median individual income, this does not lead them to have higher than its median green space access or walkability. The limitations of our study do not allow us to identify causal pathways connecting SEDH and health outcomes. Yet we demonstrate the utility of data disaggregation for addressing complex questions involving community-specific behavioral attributes and built environments with precision and rigor, which is especially useful for a diverse population.

## Figures and Tables

**Figure 1 ijerph-21-00771-f001:**
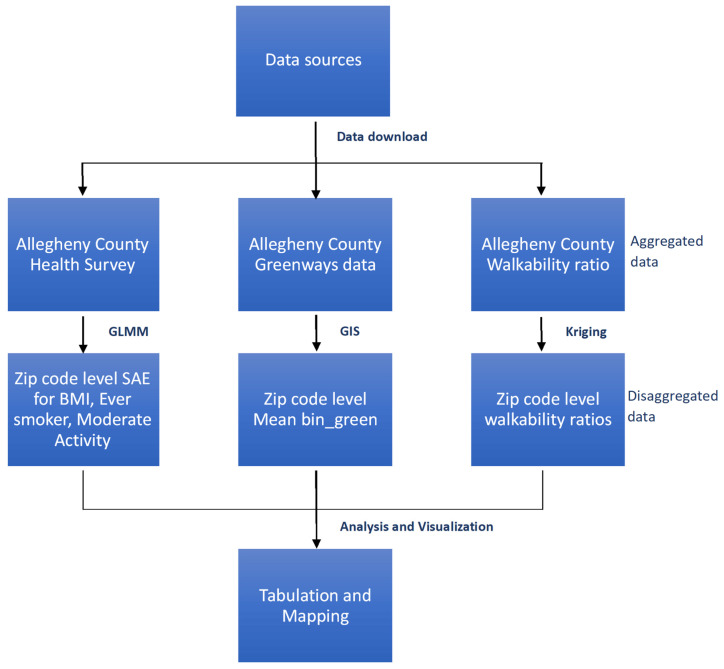
Analytical pipeline for disaggregation and use of different types of data in this study. (Abbreviations—GLMM: Generalized Linear Mixed Model; GIS: Geographic Information System; SAE: small area estimates; BMI: Body Mass Index; *Mean bin_green*: proportion of the population residing in Z that has access to any green space within 500 m).

**Figure 2 ijerph-21-00771-f002:**
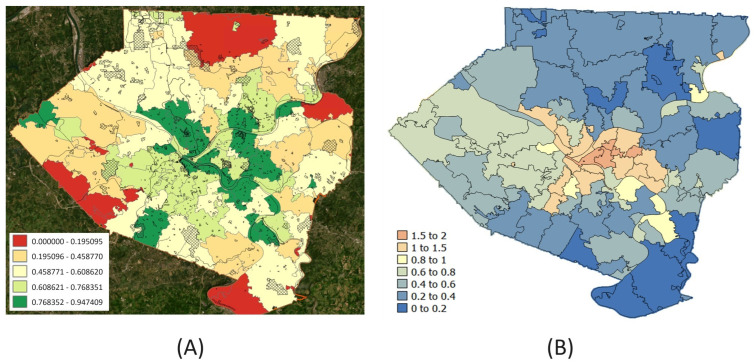
(**A**) Estimated *Mean bin_green* values for each zip code in Allegheny County. *Mean bin_green* of a zip code *Z* is defined by the proportion of the population residing in *Z* that has access to any green space within 500 m. (**B**) Walkability ratios of zip codes in Allegheny County estimated by Area-To-Point (ATP) kriging.

**Figure 3 ijerph-21-00771-f003:**
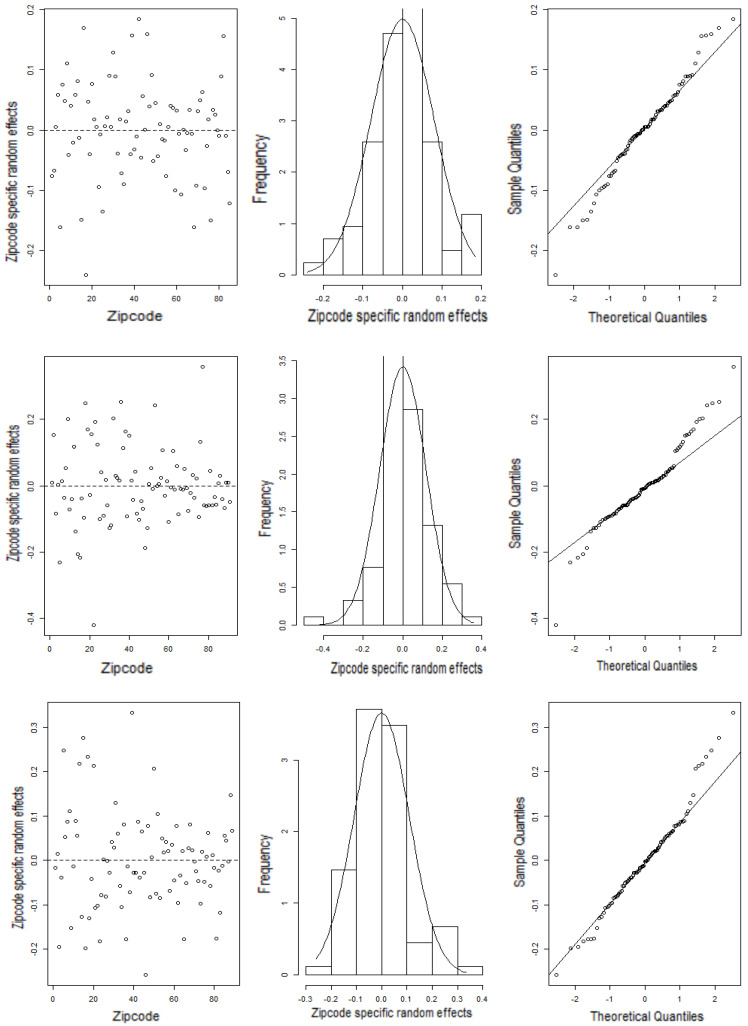
Zip code level residuals (**left**), histograms (**center**), and normal q–q plots of the zip code level residuals (**right**) for BMI, Ever-smoker, and Moderate activity (**top** to **bottom**). See definitions of the variables in Section 2.1.

**Figure 4 ijerph-21-00771-f004:**
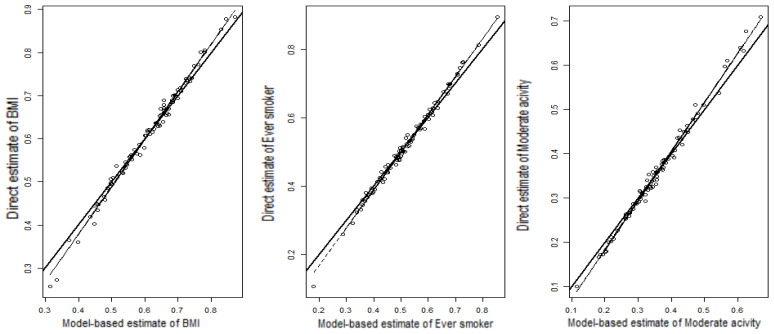
Bias diagnostic plot for BMI, Ever-smoker, and Moderate activity of zip code-wise Direct (y-axis) and Model-based (x-axis) estimates. See definitions of the variables in Section 2.1.

**Figure 5 ijerph-21-00771-f005:**
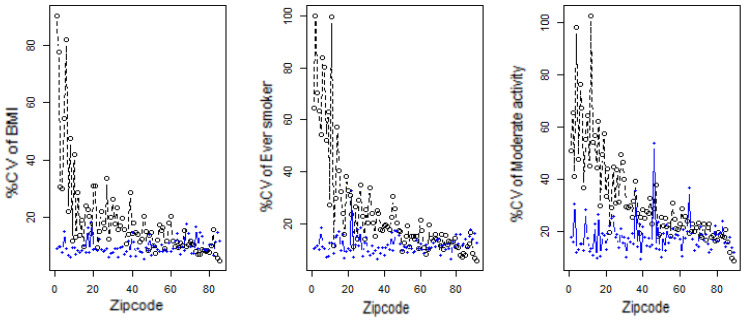
Percent CV (y-axis) for zip codes (x-axis) of Direct (in black) and small area estimates (in blue) of BMI (**left**), Ever-smoker (**center**), and Moderate activity (**right**). Zip codes are arranged in increasing order of sample size. See definitions of the variables in Section 2.1.

**Figure 6 ijerph-21-00771-f006:**
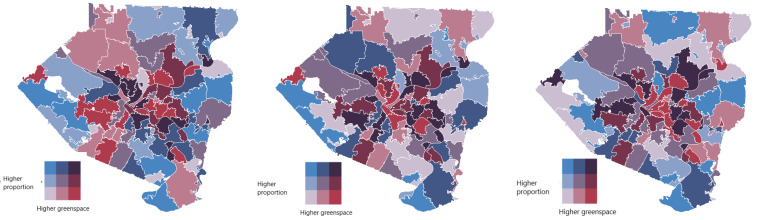
Bivariate maps show the spatial distributions of Mean bin_ green values (**top** panel) and Mean walkability ratios (**bottom** panel) against the small area estimates of BMI (**left** panel), Ever-smoker (**middle** panel), and Moderate activity (**right** panel) for Allegheny County. Increasing severity of greenspace and walkability are shown in darker shades in x- and y-directions, respectively. The legends show increasing severity of environmental and behavioral factors in darker shades in x- and y-directions, respectively. The shades used in the figure represent high, medium, and low values of the estimates. *Mean bin_green* of a zip code *Z* is defined by the proportion of the population residing in *Z* that has access to any green space within 500 m.

**Figure 7 ijerph-21-00771-f007:**
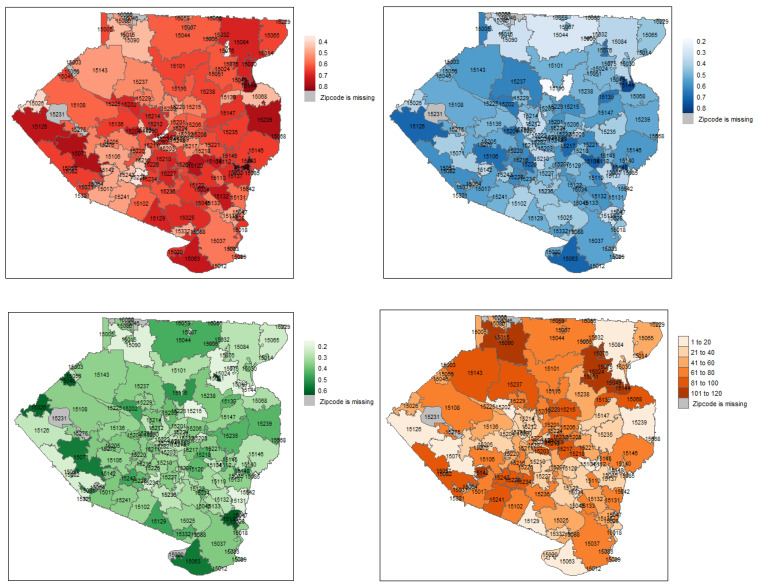
Spatial mapping of zip code level distributions of BMI (red), Ever-smoker (blue) and Moderate activity (green) estimates for Allegheny County. The median rank (orange) of each zip code based on these 3 estimates are shown in the final map, with darker shades indicating lower ranks.

**Table 1 ijerph-21-00771-t001:** Summary of the distribution of CV (in %) for the direct, and small area estimates for the three target variables $y_{1}=$ BMI, $y_{2}=$ Ever-smoker and $y_{3}=$ Moderate activity.

Values	Direct Estimate			Small Area Estimate		
	$y_{1}$	$y_{2}$	$y_{3}$	$y_{1}$	$y_{2}$	$y_{3}$
Minimum	4.93	5.99	8.98	5.33	6.63	9.68
$Q_{1}$	10.18	12.97	20.11	7.95	10.02	14.15
Median	14.49	17.37	25.46	8.94	11.37	17.39
Mean	18.64	24.89	32.22	9.63	11.84	18.18
$Q_{3}$	20.18	26.26	39.98	10.30	13.47	19.50
maximum	90.20	100.01	102.61	18.31	32.82	53.99

**Table 2 ijerph-21-00771-t002:** Local characteristics and green space access and walkability of the top 10 zip codes (with at least 100 housing units) in Allegheny County by Median Rank based on small area estimates of behavioral risk factors. Zip code data are due to www.unitedstateszipcodes.org (accessed on 1 January 2023) (Abbr.: ZIP—zip code; Lat—latitude; Long—longitude; LA—land area in sq. mile; Pop—population size; Den—population density in per sq. mile; HU—number of housing units; MI—median income in US$; MWR—mean walkability ratio; MBG—*Mean bin_green* of a zip code *Z* as defined by the proportion of the population residing in *Z* that has access to any green space within 500 m).

ZIP	Lat	Long	LA	Pop	Den	HU	MI	MWR	MBG
15035	40.39	−79.81	0.41	2129	5131	1155	40,104	0.559716	0.63831
15071	40.41	−80.19	18.32	9956	543	4402	60,172	0.455995	0.453477
15126	40.46	−80.28	22.14	7014	317	3113	66,209	0.613059	0.314695
15148	40.38	−79.79	1.12	2814	2522	1554	26,506	0.528861	0.563906
15104	40.40	−79.86	2.45	9038	3691	4845	23,403	1.027945	0.765937
15112	40.40	−79.84	0.84	3292	3937	1890	40,431	0.711146	0.762335
15084	40.61	−79.81	27.82	10,130	364	5005	44,295	0.221643	0.502483
15129	40.3	−80.00	8.05	10,920	1356	4629	64,484	0.133422	0.534197
15202	40.50	−80.08	4.40	19,685	4477	10448	43,791	1.009522	0.786843
15239	40.48	−79.74	15.98	21,024	1315	8746	65,750	0.085331	0.346972

## Data Availability

The data presented in this study are available on request from the corresponding author.

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
