# Peer review of "Disaggregation of Green Space Access, Walkability, and Behavioral Risk Factor Data for Precise Estimation of Local Population Characteristics"

_ijerph, 2024, doi:10.3390/ijerph21060771_

Round 1
Reviewer 1 Report
Comments and Suggestions for Authors
This article took the zip code area as the basic spatial carrier and discussed the estimation methods for the green space access, the walkability, and the behavioral risk factors in each spatial unit. In particular, the estimation of behavioral risk factors was analyzed through in-depth statistical testing. Despite having solid sampling survey data as the research foundation, this study did not integrate the estimation results of the three sections well or responded to the preset goal of "precise estimation of population characteristics" in the title.
Specifically, there has not been sufficient analysis and discussion on the estimation of green space access and walkability. Furthermore, this article does not present the connection between these works and the assessment of behavioral risk factors. The above issues create a strong sense of disconnection in the results and the discussion section.
The authors are suggested to modify this article from the following aspects:
1. Reorganize the relationship between the three parts of the methodology. It is best to add a flowchart to clarify the sequence of several estimations and their mutual relationships.
2. In the Results, it is recommended to follow the sorted methodology and reproduce the content in sections. They should be split into different sections. The textual descriptions and the figures should be matched to improve the legibility. In addition, there are resolution issues with the text in many figures. Please consider modifications for Figures 3, 4, 5, and 6.
3.In the Discussion, more valuable reviews or reflections should be provided on the specific statistical testing results and indicator linkage analysis. The authors should notice that almost all of the references in Discussion are from 2015 or even before 2010, which can easily lead to doubts about their timeliness. It is suggested to adjust the bibliography.
Reviewer 2 Report
Comments and Suggestions for Authors
Reviewer 3 Report
Comments and Suggestions for Authors
I have uploaded my observations and comments for the authors in the file attached.

Reviewer 4 Report
Comments and Suggestions for Authors
Thank you very much for sending the paper to review. In my opinion, the topic is very interesting and the content presents the current data about the "hot" topic of active transport, namely walkability, as well as green spaces and built environment to secure the high quality space for local space users.
I evaluate the structure and content high. All the needed constructs were addressed before the description of method. The aim is clearly stated and addressed in the conlcusion and earlier, in the results. Then, the method and materials are described in detail and all the necessary data are mentioned in the text. The results are stuctured well and in my opinion, refer to the results promised in the abstract and introduction.
I have only some minor comments - please find the comments in order of appearance in the text:
1. Abstract: Please be more focused on the contribution and impact of the results.
2. If there should be SDEH, and not the SEDH? According to the full wording? (juest to be considered by authors)
3. Please correct the 3.1. heading
Round 2
Reviewer 1 Report
Comments and Suggestions for Authors
The authors have made many commendable improvements to the manuscript. Overall, the estimation of behavioral risk factors for residents in zip code is very interesting, with complete logic, clear methods, and convincing results. However, the other two estimation is slightly weak. In Result and Discussion, although visualization was carried out (Figure 6) and a large amount of additional information was introduced (Table 2), the discussion was superficial.
There are three questions for the authors:
1. Is there primary and secondary differences among the three estimations conducted in the article?
2. If there is a difference. Is it necessary to estimate the accessibility and walkability of green spaces in the logic of the article? Do they provide any help or support for estimating the behavioral risk factors?
3. If there is no difference. Do the proportions of the three estimated results need to be adjusted? In addition, the Result section in the manuscript was not segmented as stated.
Author Response
The authors have made many commendable improvements to the manuscript. Overall, the estimation of behavioral risk factors for residents in zip code is very interesting, with complete logic, clear methods, and convincing results. However, the other two estimation is slightly weak. In Result and Discussion, although
visualization was carried out (Figure 6) and a large amount of additional information was introduced (Table 2), the discussion was superficial.
Comment 1. Is there primary and secondary differences among the three estimations conducted in the article?
Authors' reply: The primary differences among the estimations are due to the distinctive spatial approaches of the three estimation techniques used. Each of these techniques are well suited for the kind of datasets to which they were applied – SAE for survey data (ACHS), GIS for geospatial data (green spaces), and kriging for geostatistical data (walkability ratios). For all three approaches, we have computed local estimates and associated variance measures. Thus, we used the rankings based on the said spatial estimates, which allowed us to compare the zip codes using spatial maps and associated tables.
Comment 2. If there is a difference. Is it necessary to estimate the accessibility and walkability of green spaces in the logic of the article? Do they provide any help or support for estimating the behavioral risk factors?
Authors' reply: Any differences, which may be due to different estimation approaches, would be mitigated by the relative ordering of the zip codes as obtained by their respective ranks. The small area estimation of behavioral risk factors was done with the help of census-based covariates (and not green space access or walkability). As noted in the discussion, we chose to avoid building a model that tries to explain behavior in terms of green space access and walkability, and instead compared the zip codes spatially based on their ranks in these variables. Testing for possible causal association among these variables is beyond the scope of our study.
Comment 3. If there is no difference. Do the proportions of the three estimated results need to be adjusted? In addition, the Result section in the manuscript was not segmented as stated.
Authors' reply: We do not think this is necessary due to point #2 above.
